# The Spatial Heterogeneity Effect of Green Finance Development on Carbon Emissions

**DOI:** 10.3390/e24081042

**Published:** 2022-07-29

**Authors:** Langang Feng, Shu Shang, Sufang An, Wenli Yang

**Affiliations:** 1Guizhou Key Laboratory of Big Data Statistics and Analysis, Guizhou University of Finance and Economics, Guiyang 550025, China; fenglangang@mail.gufe.edu.cn; 2School of Big Data Application and Economics, Guizhou University of Finance and Economics, Guiyang 550025, China; shushang1997@gmail.com (S.S.); wenliyang@mail.gufe.edu.cn (W.Y.); 3College of Information and Engineering, Hebei GEO University, Shijiazhuang 050031, China; 4Intelligent Sensor Network Engineering Research Center of Hebei Province, Shijiazhuang 050031, China

**Keywords:** green finance, carbon emissions, entropy method, spatial Durbin model, heterogeneity

## Abstract

This paper uses the entropy method to estimate China’s green financial development from four aspects, namely, green credit, green securities, green insurance, and green investment, based on the provincial-level panel data from 2008 to 2019. The spatial Durbin model (SDM) is adopted to estimate the spatial effect of green finance on carbon emissions. We then compare the heterogeneous effect in the South and North of China. The results show that China’s green financial development can significantly reduce carbon emissions, and regional heterogeneities are obvious. In the South of China, this effect from local and adjacent regions is not significant, while on the whole, green finance can significantly reduce carbon emissions; but for Northern China, this effect is not significant; nationally, the development of green finance and carbon emissions in adjacent areas showed an inverted U-shaped relationship. China’s green financial development and carbon emissions also showed an inverted U-shaped relationship. These results suggest that the effect of green finance development on carbon emissions exhibits substantial regional heterogeneity in China. Our paper provides some concrete empirical evidence for policymakers to formulate green financial policies to achieve the double carbon goal in China.

## 1. Introduction

Since the 1978 reform and opening-up policy, China has made remarkable progress in economic growth, industrialization, and social development. However, in the process, resource consumption has become excessive, and the environment has deteriorated [1]. According to World Bank statistics (2019), China, the European Union, and the United States are the top three carbon emitters in the world. The total carbon emissions of these three countries account for more than half of all global emissions. Meanwhile, China’s carbon emissions exceed the combined carbon emissions of the European Union and the United States. Figure 1 shows that China’s carbon emissions steadily increased from 2008 to 2019 (Figure 1 was drawn by software Origin 2018). Therefore, China must protect the environment [2]. In the 2011 National People’s Congress and the Chinese People’s Political Consultative Conference (NPC and CPPCC), emission peak and carbon neutrality were included in the report on the Work of the Government for the first time and were included in China’s 14th Five-Year Plan. Hence, it is necessary and urgent to control carbon emissions. However, policies mandatorily reducing carbon emissions from polluting industries have many limitations. The Chinese government uses financial tools to reduce carbon emissions [3] and develop green finance. 

In 2015, green finance was listed in China’s major national policy for the first time. In 2019, the development of green finance in China transformed from quantity to quality. Researchers are also paying attention to green finance development [4]. Recent studies focus on theory, practice, and economic growth and investigate the development of green finance in China from various aspects [5], such as policies, structure, and environmental benefits [5]. With the deepening development of green finance, researchers examine the effect of green finance on environmental protection [4]. Some studies point out that when green finance is effective, this improved financial development leads to higher environmental quality [6]. They also show that green finance can promote technological progress, reduce pollutant emissions, and is an important factor in improving the regional environment [7].

The recent papers studying green finance, in general, adopt a single indicator [8] and use empirical analysis, such as auto-regression, which does not consider spatial factors. A few researchers adopt spatial lag and error models to examine the relationship between green finance and the environment [9]. However, these methods cannot explain the spatial spillover effect of neighboring areas. Furthermore, studies examining the impacts of green finance on the environment generally treat the environment as a whole [10]. Few studies specifically investigate how green finance affects carbon emissions or examine the regional differences between the north and south of China. Meanwhile, carbon emissions reduction is mandatory for green finance in the direction of China’s economic development [2]. It is important to understand green finance’s effect on carbon emissions and its spatial heterogeneity. 

Based on the Qinling–Huaihe Line, Northern and Southern of China have great differences levels of heating and economic development [11]. Carbon emissions and financial development are also fundamentally different between the Northern and Southern China. Understanding the heterogeneity of the impacts of green finance on carbon emissions in Northern and Southern China would be helpful for designing effective policies regarding green finance to reduce carbon emissions.

Our paper makes a threefold contribution to the literature. First, we construct an index to measure green finance. We combine official documents and relevant research and adopt the entropy method to construct a green finance development index. Second, we employ the spatial Durbin model (SDM) to estimate the spatial spillover effect of green finance on carbon emissions. Third, we emphasize the heterogeneous regional effects. This paper not only discusses the effect of green finance on carbon emissions at the national level, but also analyzes the regional differences between Northern and Southern China.

## 2. Empirical Strategy

### 2.1. The Entropy Method

The index is generally adopted to measure the development level of green finance [2]. The selection of weight determines the index. Currently, the main construction methods include the entropy method [2], the analytic hierarchy process [12], and the principal component analysis [13]. The analytic hierarchy process subjectively assigns weights, which may undermine the objectivity of the results. Principal component analysis is required reduce the dimension of the index, but the principal component becomes difficult to explain after dimension reduction.

Therefore, we use the entropy method to construct the green financial index. The entropy method determines the weights according to the number of information indicators provided. The weight is heavier when the indicator provides more information and has a larger variation [2]. The entropy method originated from the information theory. Information entropy can measure the degree of the dispersion of indicators. However, the original entropy method does not take time into account. Following Chen and Chen (2021) [2], in this paper, we incorporate time into the entropy method. The improved entropy method model is as follows:

First, the data should be standardized.

There are positive and negative sub-indicators of green finance. In order to combine different indicators, data should be standardized, and the formula for the standardization of positive indicators is as follows: (1)$$f_{j}\left(it\right)=\frac{x_{j}\left(it\right)-min{\;}_{1\;\leq \;i\;\leq \;n}min{\;}_{1\;\leq \;t\;\leq \;T}\left[x_{j}\left(it\right)\right]}{max_{1\;\leq \;i\;\leq \;n}max_{1\;\leq \;t\;\leq \;T}\left[x_{j}\left(it\right)\right]-min_{1\;\leq \;i\;\leq \;n}min{\;}_{1\;\leq \;t\;\leq \;T}\left[x_{j}\left(it\right)\right]}$$

The formula for the standardization of negative indicators is as follows:(2)$$f_{j}\left(it\right)=\frac{min_{1\leq i\leq n}min_{1\leq t\leq T}\left[x_{j}\left(it\right)\right]-x_{j}\left(it\right)}{max_{1\;\leq \;i\;\leq \;n}max{\;}_{1\;\leq \;t\;\leq \;T}\left[x_{j}\left(it\right)\right]-min_{1\;\leq \;i\;\leq \;n}min_{1\;\leq \;t\;\leq \;T}\left[x_{j}\left(it\right)\right]}$$
where j represents the index, i represents the region, t represents the year, $f_{j}\left(it\right)$ represents the standardized value of the region’s first index in the first year, and $x_{j}\left(it\right)$ represents the original value of the region’s first index in the first year; max, min represent the maximum and minimum value of the j index; T represents the number of years, and n represents the number of regions. 

Second, determine the contribution degree of indicators.
(3)$$y_{j}\left(it\right)=\frac{f_{j}\left(it\right)}{\sum_{t=1}^{T}\sum_{i=1}^{n}f_{j}\left(it\right)}$$
where $y_{j}\left(it\right)$ represents the contribution of indicator j in area i in year t. 

Third, calculate the entropy of the j index.
(4)$$e_{j}=-\frac{1}{k}\sum_{t=1}^{T}\overset{n}{\underset{i=1}{\sum }}y_{i}\left(it\right)lny_{i}\left(it\right)$$

In the formula, *k* > 0, i represents the number of regions, and t represents the number of years.

Fourth, calculate the information utility value of indicator j.
(5)$$g_{j}=1-e_{j}$$

Fifth, according to the information utility value to calculate the weight of each index:(6)$$w_{j}=\frac{g_{j}}{\sum_{j=1}^{m\;}g_{j}}$$

Sixth, the green finance development index GF is calculated by weighting multiple linear functions:(7)$$GF_{j}=\overset{m}{\underset{j=1}{\sum }}g_{j}\times y_{j}\left(it\right)$$

### 2.2. Model Specification

In order to examine the impact of green finance on carbon emissions, we use the software Stata16.0 to conduct regression estimation of the following SDM model, and the SDM model is designed as follows:(8)$$lnCO_{2}=\rho W_{k}lnCO_{2}+\beta_{1}lnGF+\beta_{2}Z_{it}+\theta_{1}W_{k}lnGF+\theta_{2}W_{k}Z_{i}+v_{i}+v_{t}+\varepsilon_{it}$$
(9)$$\mathrm{where}\;\varepsilon_{it}=\lambda W_{k}\varepsilon_{it}+\gamma_{it}$$

$LnCO_{2}$ is the explained variable representing the carbon emission intensity, the core explanatory variable LnGF represents the development level of green finance, and Z is the control variable. *W* is a spatial weight matrix, $\beta_{1},\;\beta_{2}$ represents the effect of explanatory variables on local carbon emission intensity, $\theta_{1},\;\theta_{2}$ is a spatial coefficient, and g represents the effect of explanatory variables on carbon emission intensity in other regions. ρ is space self-regression coefficient, $v_{i}$ is space fixed effect, $v_{t}$ is time fixed effect, and $\varepsilon_{it}$ is error term.

### 2.3. Data

#### 2.3.1. Dependent Variable

Our dependent variable Ln CO_2_ is the carbon emission intensity, which is measured as the logarithm of (CO_2_/real GDP). The dependent variable represents the logarithm of the amount of CO_2_ emitted when one unit of GDP is produced. The carbon emission intensity is controlled for economic scale, reflects the CO_2_ emission per unit of output, and can represent the quality and efficiency of economic development.

#### 2.3.2. Regressor of Main Interest

Since green finance is a relatively new concept in China, researchers have not yet come to an agreement on the measurement of green finance. The recent measurement relies on the construction of the index system. Following Liu et al. (2021) [14] and Liu et al. (2019) [15], the index system is categorized into different types of financial services, namely, green credit, green insurance, green securities, and green investment. The index system representing the provincial-level development of green finance is constructed according to these four dimensions. The current research on the measurement of green finance development mainly starts from the perspectives of capital supply, capital demand, and the characteristics of green finance. From the first perspective, the development level of regional green finance is measured from the capital supply side. For example, Zhou et al. (2022) measure the development of green finance from the supply side of green credit, green insurance, green investment, and carbon finance [4]. From the second perspective, starting from the capital demand side, the financial data of green-listed companies are used for comprehensive analysis to characterize the development of green finance [16]. The third perspective uses economic, financial, and environmental dimensions of China’s provincial green financial development index as measurements [17].

As mentioned above [4,16,17], we start from the supply side of green finance funds. In order to reflect green finance more comprehensively, according to the Guidelines for Establishing the Green Financial System, issued by the People’s Bank of China, and referring to the indicators selected by Zhou et al. (2022) [4], Chen and Chen (2021) [2], Liu et al. (2021) [14], Liu et al. (2019) [15], and other scholars, the development level of green finance is defined from four aspects: green credit, green insurance, green securities, and green investment. Specifically:

In terms of green credit, the proportion of energy-saving and environmental protection loans and the proportion of interest in high energy-consuming industries are used. Since green credit policy can support the development of the environmental protection industry, it is through bank credit that new energy and clean industries adjust their industrial structure. Therefore, energy saving and environmental protection loans are used to represent the GDP’s proportion of energy saving and environmental protection loans. According to the high energy consumption industry classification standard formulated by the National Development and Reform Commission (the classification standard of high energy-consuming industries, formulated by the National Development and Reform Commission, selects six high energy-consuming industries—the electricity, thermal production, and supply industry; ferrous metal smelting and rolling processing; non-ferrous metal smelting and rolling processing; the non-metallic mineral products industry; the petroleum processing, coking, and nuclear fuel processing industry; and the chemical raw materials and chemical products manufacturing industry), the high energy consumption industry is selected. The ratio of six high energy-consumption industrial interest expenditures to the total industrial interest expenditure is used to reflect the development level of green credit [4]. In the current industry, loan interest rate difference is small, the industry interest expenditure is mainly related to the loan scale, and the proportion of interest expenditure indirectly reflects the proportion of the loan scale [4]. High energy-consuming industries generally have the characteristics of high pollution and high consumption, which are the key areas of national development restriction in recent years. The interest proportion of high energy-consuming industries can reflect the strength of banks to curb the further deterioration of the resource environment due to pollution. Therefore, high-energy industrial interest rates are a reverse indicator of green credit [15,18].

In terms of green insurance, the proportion of the agricultural insurance scale and the compensation rate are used to approximately reflect the development of green insurance. The agricultural insurance scale is represented by the proportion of the agricultural insurance expenditure and income. The agricultural compensation rate is represented by the total agricultural insurance expenditure proportion [4,19]. Green insurance uses environmental liability insurance to measure the most appropriate rate. However, China’s environmental pollution liability insurance is still in its infancy and lacks authoritative statistical data and information. Agriculture is the industry most affected by the natural environment. It has the highest correlation with natural environmental protection and carbon emission reduction and has high public attributes. These factors are highly similar to environmental liability insurance, which can reflect the characteristics of green insurance to a certain extent. Therefore, the scale ratio of agricultural insurance and the compensation rate is used to measure the level of green insurance development [4,20].

In terms of green securities, referring to the research of scholars [2], the market value of the green industry is represented by the market value of the green finance industry (excluding ST plate). Through the proportion of green industry market value in A shares, the level of green industry market value in the total market value of A shares is reflected. Green industry aims to adopt new technologies and processes, such as cleaner production, to achieve the goal of high output and low pollution. The transformation of traditional industries to green industries can reduce pollutants, especially carbon dioxide emissions, so as to achieve the goal of environmental protection; therefore, we use the market value ratio of the green industry to measure the level of green investment [21].

In terms of green investment, the proportion of energy-saving and environmental protection expenditure and pollution control investment are used to represent green investment [2,4] to reflect the financing level of green industry through channels other than bank credit and the capital market [2,4]. Public expenditure for energy conservation and environmental protection mainly reflect the expenditure level of energy conservation and environmental protection in public finance. Since energy conservation and environmental protection are typically public goods, this field is a key area of public investment. In particular, due to the lack of ongoing effective direct investment and financing channels in China, investment in energy conservation and environmental protection is more dependent on public finance. Therefore, the inclusion of public investment in green investment evaluation is in line with the actual situation in China and can comprehensively reflect the development level of green investment [2,4].

Based on this, this paper uses the entropy method to construct the index system of green credit, green investment, green insurance, green securities, and green finance. Figure 2 shows the changes in green credit, green securities, green insurance, green investment, and green financial indicators from 2008 to 2019. From 2008 to 2019, except for a slight decline in the green credit index, the green credit indicators showed a fluctuating growth trend. The figure shows that although green credit has declined slightly, green finance, green insurance, green securities, and green investment have shown a fluctuating growth trend. China’s emphasis on green finance is gradually increasing.

#### 2.3.3. Other Control Variables 

There are great differences in the level of economic development, energy utilization, economic structure, environmental regulation, R&D investment, and opening up in different regions of China, which makes the effect of green financial development on carbon emissions exhibit heterogeneity in different regions [22]. The variables selected in the existing research are different. Some scholars consider the level of economic development, trade openness, foreign direct investment, human capital, and financial development as factors affecting carbon emissions [23]. Some scholars regard globalization, financial development, government expenditure, and institutional quality as independent variables to study the impact of green financial development on carbon emission reduction [24]. Other scholars have demonstrated the impact of environmental regulation and technological innovation on the environment [25]. Based on the regional situation and the selection of existing research variables, nine variables are selected from five aspects, including scale, technology, structure, trade, and government regulation as the control variables of the impact of green finance on carbon emissions.

In terms of scale, this study measures the scale from the per capita GDP level and population size. The per capita GDP level is represented by the ratio of real GDP to the total population, and the total population measures the population size at the end of the year [23], in terms of technology. This study considers the proportion of R&D expenditures in the nominal GDP to measure the input of technical expenditure. Energy intensity measures the energy intensity by considering the standard coal consumption level in each GDP [25]. In terms of structure, it is characterized by the proportion of industrial structure in the secondary industry and the proportion of the urbanized population. Among them, the secondary industry is the industry with the most pollution emissions in the tertiary industry, so the ratio of the secondary industry to the total output value is used as a measure of the proportion of the secondary industry. The urbanization rate represents the proportion of the urban population and can reflect the structure of the urban population [23]. In terms of trade, we draw on the practice of existing research using two aspects, foreign direct investment and openness, including the ratio of foreign direct investment to nominal GDP and the ratio of import and export volume to real GDP [23]. In terms of government control, measuring the intensity of government regulation by the level of environmental regulation, the ratio of investment in pollution control projects to nominal GDP is used to characterize environmental regulation [25].

#### 2.3.4. Summary Statistics

The relevant data of 30 provinces, cities, and autonomous regions (except Tibet, Hong Kong, Macao, and Taiwan) in China from 2008 to 2019 (In 2008, the Industrial Bank adopted the “Equator Principles” and introduced the concept of green finance for China. Since the data are only available up to 2019, this paper selects the data from 2008 to 2019 as its research object) are used as research samples (This article samples companies that exist throughout this time period, excluding exits). Carbon emission data are from CEADS database. The data on green credit come from the China Banking Social Responsibility Report. The interest expenditure of high energy-consuming industries and the interest expenditure of industrial industries come from the China Industry Statistical Yearbook and regional statistical yearbooks over the years. The data regarding the agricultural insurance scale and compensation rate come from the Guotaian Database. The market value of green industry and the total market value of A-shares come from the Tonghuashun Database. The pollution control investment ratio data come from the China Environment Statistical Yearbook over the years. GDP, the total population at the end of the year, the proportion of the secondary industry, urbanization rate, import and export data, R&D funds, and standard coal and pollution control projects completed in each year’s investment are from the regional statistical yearbook and the China Statistical Yearbook. Exchange rate data are from the official website of the People’s Bank of China (According to the exchange rate on 1 May 2021). Considering that a small amount data for the years is not disclosed or is missing in the sample, interpolation and trend calculation are used to fill the missing values.

The descriptive statistical results of each variable are shown in Table 1. It can be seen from Table 1 that there are indeed large regional differences between carbon emission intensity and green finance development levels in 30 provinces, municipalities, and autonomous regions in China from 2008 to 2019. The difference in green finance development level is related to the country’s economic development level. The difference in carbon dioxide emission intensity is related to economic scale, energy structure technology, industrial structure, foreign trade, etc., [26].

## 3. Empirical Results

### 3.1. Spatial Correlation Test

We adopt the Moran’s I (Moran index) test to check the spatial dependence and spatial autocorrelation. The results are presented in Table 2. It can be seen from Table 2 that the carbon emission intensity coefficients are significantly indigenous at the level of 1%, indicating that there is a significant spatial autocorrelation of carbon emission intensity. Furthermore, according to the sign of the Moran’s index, the carbon emission intensity exhibits positive spatial autocorrelation in China. Thus, the spatial model is adopted in the empirical analysis.

### 3.2. The Effect of Green Finance on Carbon Emissions

The Northern and Southern regions of China are divided by the Qinling–Huaihe River, the Northern region is the area north of the Qinling–Huaihe River, and the Southern region is the area south of the Qinling–Huaihe River. The Northern region burns coal for heating in winter, while the Southern region does not. In addition, there is a big difference in the economy between the North and the South. In order to explore whether these differences will affect the effectiveness of green financial development on carbon emissions, in addition to analyzing the impact of green financial development on carbon emissions in China through the spatial Durbin model, China is divided into the Northern and Southern regions, with the Qinling–Huaihe River as the boundary, in order to comprehensively and clearly analyze the heterogeneity of the impact of green financial development on carbon emissions in the Northern and Southern regions, and to put forward corresponding policy suggestions targeted to the differences.

#### 3.2.1. The National Level

Table 3 reports the effect of green finance on carbon emissions at the national level. The results are obtained from the SDM, in which the economic weighting matrix is employed. Before the analysis, we use the combination of the LM test, SDM with fixed effects, the Hausman test, and the SDM simplified test, and follow the rules of “specific-to-general” and “general-to-specific,” and eventually find that the SDM with fixed effects is the best choice. Therefore, the SDM with fixed effects is employed to examine how green finance spatially affects carbon emission intensity.

At the national level, the direct effect of green finance on carbon emission is statistically insignificant, indicating that green finance has no significant impact on local carbon emissions. The indirect effect coefficient is −0.117 and is significant at 1%. It suggests that when green finance increases by 1% in neighboring provinces, the local carbon emissions will reduce by 0.117%. Developing green finance in neighboring provinces can significantly stimulate local carbon emission reduction. Thus, green finance should not only be developed in one region, but should also be expanded to cover other regions so that the spillover effect could be enhanced. The coefficient of the total effect is −0.125. It is significant at the 5% level, indicating that a 1% increase in green finance leads to a 0.125% decrease in carbon emissions at the national level.

In addition, the spatial Durbin model is constructed by introducing the primary and secondary factors of the green financial development coefficient. The empirical results are shown in Table 3, columns 4–6. In the direct effect, the second factor of the green financial development coefficient is negative, but not significant. The quadratic coefficient of green finance in indirect and total effects is negative, and both are significant. Therefore, there is an inverted U-shaped relationship between green financial development and the indirect and total effects of carbon emissions. That is, green financial development first promotes carbon emissions to reach the threshold and then begins to reduce carbon emissions, which is in line with the concept of the environmental Kuznets curve. This relationship may exist because green finance development promotes enterprises’ innovation investment, thereby accelerating technological progress and R&D investment. Technological progress and the substitution of new clean products can achieve the effect of carbon emission reduction. However, green finance development will expand enterprises’ production scale, thereby increasing energy demand and consumption and ultimately promoting carbon emissions. Therefore, in the early stages, the development of green finance promotes carbon emissions, but in the long run, with the further development of green finance, the development of green finance inhibits carbon emissions.

Based on the existing research, this paper analyzes the impact of investment risk on carbon emissions from the perspective of private companies [27]. Through empirical analysis, it is found that venture capital’s direct and total effects on carbon emissions are not significant, but the indirect effect is significantly negative. It shows that increasing risk investment in neighboring areas can reduce carbon emissions in the region. The reason may be that the implementation scope of neighboring regions is larger, and the implementation effect is therefore greater. Thus, in order to further promote the reduction effect of venture capital on carbon emissions, the scope of implementation of venture capital should be expanded, and the intensity of venture capital should be increased. Due to limited space, the empirical results are not detailed, but interested readers can contact the author to obtain them.

#### 3.2.2. Northern China

The three effects of green finance development on carbon emission intensity in Northern China are shown in Table 4. The direct effect coefficient is −0.001, the indirect effect coefficient is −0.033, and the total effect coefficient is −0.035. None of the three effects are significant. The reason may be that the development of green finance in Northern China is still in its infancy, and the level of green finance development is low, making it difficult to achieve the effect of reducing carbon emissions. Therefore, the development process of green finance should focus on strengthening the development of green finance in Northern China in order to achieve the best effect of carbon emission reduction. The control variables with significant direct and indirect effects are foreign direct investment, openness, and environmental regulation, and their direct, indirect, and total effects are the same. The total effect of per capita GDP, the proportion of secondary industry, and the urbanization rate are not significant, which may be due to the direct effect and indirect effect direction being different, and the fact that some effects offset each other. The total effect of the total population and R&D investment is similar to the indirect effect, contrary to the direct effect, indicating that the indirect effect occupies the dominant position in the total effect of the total population and R&D investment on carbon emission intensity. The total effect of energy intensity on carbon emission intensity is similar to the direct effect, indicating that the direct effect is dominant in the process of energy intensity on carbon emission intensity.

#### 3.2.3. Southern China

The three effects of green finance development on carbon emission intensity in Southern China are shown in Table 5. The coefficients of green finance’s direct and indirect effects on the southern region’s carbon emission intensity are negative and not significantly indigenous. The total effect coefficient is −0.05. Through the 10% significant indigenous level test, the total effect is negative under the combined effect of direct and indirect effects, indicating that the development of green finance will reduce carbon emissions in general. That is, carbon emissions will reduce by 0.05 units per unit, with an increase in the level of green finance development. The direct effect and indirect effect coefficients are negative but not significant. It can be seen that although green finance can significantly reduce carbon emissions at present, the direct and indirect effects of green finance on carbon emissions are not significant. Therefore, we should continue to increase the support of green finance and expand the scope of implementation of green finance in order to bring about better carbon emission reduction effects. In terms of control variables, the direct effect, indirect effect, and total effect are significant variables of per capita GDP, R&D investment, and environmental regulation. The indirect effect of per capita GDP is dominant, and the direct effect of energy intensity is dominant.

### 3.3. Robustness Checks

We use an alternative dependent variable, i.e., the carbon emission per capita, to replace *Ln CO*_2_ carbon emission intensity in Equation (7) as a robustness check. The results are summarized in Table 6. We find that all the results, including signs and significance levels, are consistent with the corresponding baseline results. Hence, we are confident in our findings.

The abnormal values in the sample may lead to abnormal regression results. In order to avoid the influence of outliers on the regression results and further eliminate the lowest and highest 1% of sample data in carbon emissions and green finance, Table 7 reports the estimation results after eliminating outliers. In China’s Northern and Southern regions, the direction of green finance on carbon emission coefficient and the level of visibility have not changed. Thus, the regression conclusion is not affected by the sample outliers.

## 4. Conclusions and Policy Recommendations

### 4.1. Conclusions

This paper adopts panel data of China’s 30 provinces from 2008 to 2019 and constructs the green finance index from four perspectives, i.e., green credit, green securities, green insurance, and green investment. This paper uses Stata16.0 software to test the regression of the SDM model to evaluate the spatial effect of green finance on carbon emissions. We further analyze the heterogeneous effect in Southern and Northern China. The empirical results obtained in this paper are as follows. Firstly, China’s green financial development can significantly reduce carbon emissions, and regional differences are obvious. Secondly, in the Southern region, the impact of green finance development on regional and adjacent carbon emissions is not significant, but generally, green finance can significantly reduce carbon emissions. In the northern region, the impact of green finance on carbon emissions is not significant. Finally, across the country, the development of green finance has an inverted U-shaped relationship with carbon emissions in surrounding areas. In general, the development of green finance in China has an inverted U-shaped relationship with carbon emissions.

### 4.2. Policy Recommendations

The development of green finance can effectively reduce carbon emissions, but there are certain differences in this effect in regards to different regions. Therefore, implementing different green finance development policies is crucial for local contexts. Based on this, we propose the following policy recommendations.

First, in order to effectively reduce carbon emissions, the implementation scope of green finance should be expanded. Green finance can significantly reduce carbon emissions at the national level. The green finance of neighboring provinces has a significant and negative effect on local carbon emissions, while the effect of local green finance on local carbon emissions is statistically insignificant. The indirect/spillover effect is more prominent than the direct effect regarding carbon emission reduction. The total effect is very close to the indirect effect. This may be due to the fact that green finance in neighboring provinces is implemented in a larger scope than in its own provinces. As a result, the larger the scope of green finance, the more carbon emissions are reduced. It indicates that green finance should be implemented nationwide and not be concentrated in one region. The effect of green finance development on carbon emissions in the whole country is first promoted and then decreased. In the initial stage of green finance development, green finance will promote carbon emissions. With the further development of green finance, green finance will reduce carbon emissions after reaching a certain threshold. There is an inverted U-shaped relationship between green finance in neighboring regions and carbon emissions in the region, and the total effect of green finance on carbon emissions is also an inverted U-shaped.

Second, in general, the development of green finance in the Southern region and across China can significantly reduce carbon emissions, while the effect of green finance in the Northern region is not significant. Compared with the South and the whole country, the effect of green finance on carbon emissions in the South and the whole country is not significant. The development of green finance in the region has a significant spatial spillover effect on carbon emissions in adjacent areas, and there is no such spatial spillover effect in the Southern region. Because of these substantial regional differences, in order to achieve the desired target, that is, emission peak and carbon neutrality, policies that stimulate green finance have to adjust to accommodate local conditions, such as heating supply, economic development level, green finance development level, the proportion of the secondary industry, population, and urbanization level.

Third, avoid “one size fits all” by implementing differentiated green finance development policies in different regions of the North and South. Since green finance can effectively reduce carbon emissions and achieve the goal of emission peak and carbon neutrality, we need to make an effort to stimulate green finance development. The prospective development of green finance is promising. However, we still need to identify and solve the difficulties and challenges in the process. One of the important aspects of green finance is information disclosure. It is necessary to require listed firms to disclose relevant environmental information and green investment policies. In addition, it is important to coordinate the relationship between government and financial institutions and provide low interest rates and low loan thresholds to financial institutions developing green finance.

## Figures and Tables

**Figure 1 entropy-24-01042-f001:**
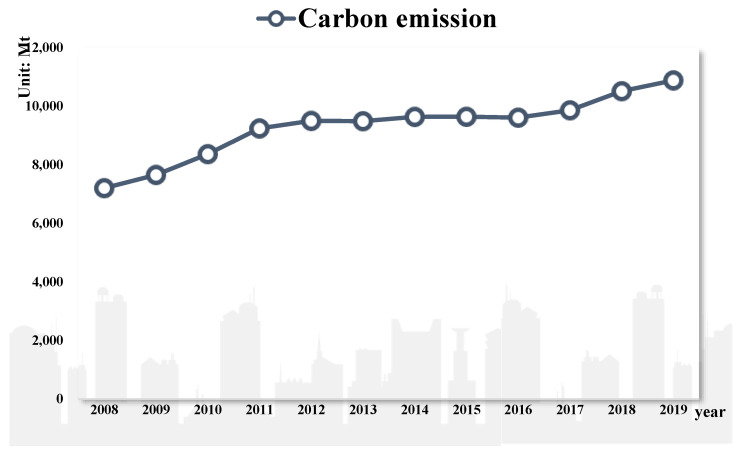
China’s carbon emissions from 2008 to 2019.

**Figure 2 entropy-24-01042-f002:**
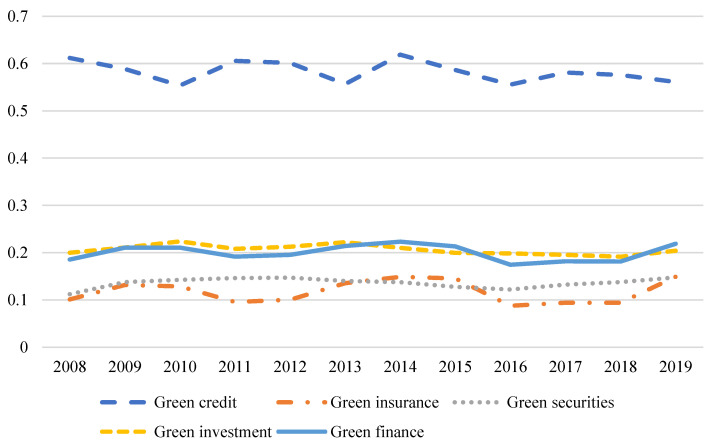
Schematic diagram of changes in green credit, green securities, green insurance, green investment, and green finance indexes.

**Table 1 entropy-24-01042-t001:** Descriptive statistical analysis results of variables, 2008 to 2019.

Variable	2008–2019				
	N	Mean	Standard Deviation	Min	Max
Ln CO_2_	360	3.168	0.598	1.374	4.796
Ln GF	360	4.604	0.658	3.364	6.770
Ln PGDP	360	3.354	0.520	2.050	4.805
Ln POP	360	8.193	0.739	6.317	9.352
Ln IND	360	3.785	0.230	2.785	4.119
Ln URB	360	3.990	0.228	3.371	4.495
Ln FDI	360	9.421	1.209	4.655	11.323
Ln OPEN	360	11.889	1.121	7.507	14.274
Ln R&D	360	6.974	1.252	0.764	9.376
Ln ENG	360	4.732	0.508	3.496	5.965
Ln ER	360	11.452	0.831	7.633	13.807

Note: Variables are in the form of natural logarithms.

**Table 2 entropy-24-01042-t002:** Moran’s index of CO_2_ emission intensity in China, 2008–2019.

Year	Moran Index	*Z* Value	Year	Moran Index	*Z* Value
2008	0.414 ***	3.718	2014	0.380 ***	3.457
2009	0.398 ***	3.589	2015	0.353 ***	3.230
2010	0.419 ***	3.858	2016	0.351 ***	3.210
2011	0.382 ***	3.507	2017	0.332 ***	3.076
2012	0.407 ***	3.686	2018	0.304 ***	2.831
2013	0.368 ***	3.356	2019	0.303 ***	2.829

Note: The significance levels 1% is noted by ***.

**Table 3 entropy-24-01042-t003:** Three effects of green finance on carbon emissions in China.

Variable	(1) Direct Effect	(2) Indirect Effect	(3) Total Effect	(4) Direct Effect	(5) Indirect Effect	(6) Total Effect
Ln GF	−0.008	−0.117 ***	−0.125 **	0.136	0.692 *	0.829 *
	(0.017)	(0.040)	(0.051)	(0.136)	(0.367)	(0.432)
Ln GF * Ln GF				−0.016	−0.088 **	−0.104 **
				(0.015)	(0.040)	(0.047)
Ln PGDP	−0.271 ***	0.270 **	−0.001	−0.259 ***	0.318 ***	0.059
	(0.044)	(0.119)	(0.124)	(0.044)	(0.113)	(0.120)
Ln POP	0.112 ***	−0.067	0.045	0.113 ***	−0.081 *	0.032
	(0.020)	(0.052)	(0.061)	(0.020)	(0.049)	(0.059)
Ln IND	0.079	−0.230 *	−0.151	0.047	−0.464 ***	−0.417 **
	(0.049)	(0.134)	(0.154)	(0.056)	(0.174)	(0.203)
Ln URB	0.019	1.324 ***	1.343 ***	0.027	1.288 ***	1.315 ***
	(0.103)	(0.301)	(0.320)	(0.099)	(0.282)	(0.296)
Ln FDI	0.018 *	−0.018	0.000	0.019 *	−0.005	0.014
	(0.011)	(0.029)	(0.035)	(0.011)	(0.029)	(0.034)
Ln OPEN	0.039 **	−0.239 ***	−0.200 ***	0.042 ***	−0.237 ***	−0.195 ***
	(0.016)	(0.042)	(0.050)	(0.008)	(0.040)	(0.046)
Ln R&D	−0.045 ***	0.079 ***	0.035	−0.042 ***	0.080 ***	0.038
	(0.008)	(0.025)	(0.029)	(0.008)	(0.025)	(0.029)
Ln ENG	1.205 ***	−0.391 ***	0.814 ***	1.221 ***	−0.366 ***	0.855 ***
	(0.038)	(0.115)	(0.131)	(0.037)	(0.107)	(0.118)
Ln ER	0.074 ***	0.118 ***	0.192 ***	0.070 ***	0.104 ***	0.174 ***
	(0.013)	(0.037)	(0.042)	(0.012)	(0.035)	(0.040)

Notes: Robust standard errors are in parentheses. The significance levels 10%, 5%, and 1% are noted by *, **, and ***, respectively.

**Table 4 entropy-24-01042-t004:** Three types of effects of green finance on carbon emissions in Northern China.

Variable	Direct Effect	Indirect Effect	Total Effect	Variable	Direct Effect	Indirect Effect	Total Effect
Ln GF	−0.001	−0.033	−0.035	Ln FDI	0.067 ***	0.153 ***	0.220 ***
	(0.024)	(0.056)	(0.072)		(0.016)	(0.041)	(0.049)
Ln PGDP	0.377 ***	−0.340	0.037	Ln OPEN	−0.048 **	−0.360 ***	−0.409 ***
	(0.096)	(0.305)	(0.351)		(0.022)	(0.057)	(0.068)
Ln POP	0.025	−0.200 **	−0.176	Ln R&D	−0.006	0.152 ***	0.146 ***
	(0.031)	(0.092)	(0.106)		(0.011)	(0.036)	(0.043)
Ln IND	0.255 **	−0.451	−0.196	Ln ENG	1.175 ***	−0.034	1.141 ***
	(0.121)	(0.334)	(0.432)		(0.053)	(0.202)	(0.233)
Ln URB	−0.668 ***	1.687 ***	1.019	Ln ER	0.074 ***	0.139 **	0.213 ***
	(0.159)	(0.455)	(0.487)		(0.020)	(0.054)	(0.067)

Notes: Robust standard errors are in parentheses. The significance levels 10%, 5%, and 1% are noted by **, and ***, respectively.

**Table 5 entropy-24-01042-t005:** Three types of effects of green finance on carbon emissions in Southern China.

Variable	Direct Effect	Indirect Effect	Total Effect	Variable	Direct Effect	Indirect Effect	Total Effect
Ln GF	−0.021	−0.029	−0.050 *	Ln FDI	0.016	0.145 ***	0.161 ***
	(0.016)	(0.022)	(0.030)		(0.014)	(0.029)	(0.035)
Ln PGDP	−0.431 ***	0.680 ***	0.248 **	Ln OPEN	0.020	−0.039	−0.019
	(0.095)	(0.097)	(0.098)		(0.016)	(0.033)	(0.034)
Ln POP	0.075 **	−0.078	−0.003	Ln R&D	−0.048 ***	−0.092 **	−0.140 ***
	(0.038)	(0.131)	(0.153)		(0.015)	(0.038)	(0.040)
Ln IND	0.177 ***	−0.362 **	−0.185	Ln ENG	1.059 ***	−0.517 ***	0.542 ***
	(0.061)	(0.170)	(0.163)		(0.068)	(0.115)	(0.135)
Ln URB	0.151	−1.054 ***	−0.903 ***	Ln ER	0.010	−0.021	−0.011
	(0.174)	(0.291)	(0.262)		(0.013)	(0.027)	(0.028)

Notes: Robust standard errors are in parentheses. The significance levels 10%, 5%, and 1% are noted by *, **, and ***, respectively.

**Table 6 entropy-24-01042-t006:** Robustness test for substitution of explained variables.

Variable	China			Northern China			Southern China		
	Dependent Variable: Ln PCO_2_			Dependent Variable: Ln PCO_2_			Dependent Variable: Ln PCO_2_		
	Direct Effect	Indirect Effect	Total Effect	Direct Effect	Indirect Effect	Total Effect	Direct Effect	Indirect Effect	Total Effect
Ln GF	−0.008	−0.117 ***	−0.125 **	−0.001	−0.033	−0.035	−0.021	−0.029	−0.050 *
	(0.016)	(0.040)	(0.051)	(0.024)	(0.056)	(0.072)	(0.016)	(0.022)	(0.030)
Ln PGDP	0.728 ***	0.267 **	0.995 ***	0.377 ***	−0.340	0.037	0.568 ***	0.677 ***	1.245 ***
	(0.045)	(0.116)	(0.121)	(0.096)	(0.305)	(0.351)	(0.097)	(0.095)	(0.097)
Ln POP	0.112 ***	−0.068	0.044	0.025	−0.200 **	−0.176 *	0.075 *	−0.078	−0.003
	(0.020)	(0.052)	(0.061)	(0.031)	(0.092)	(0.106)	(0.038)	(0.132)	(0.154)
Ln IND	0.078	−0.231 *	−0.153	0.255 **	−0.451	−0.196	0.177 ***	−0.363 **	−0.185
	(0.048)	(0.135)	(0.155)	(0.121)	(0.334)	(0.432)	(0.061)	(0.171)	(0.164)
Ln URB	0.020	1.326 ***	1.345 ***	−0.668 ***	1.687 ***	1.019 **	0.152	−1.052 ***	−0.900 ***
	(0.103)	(0.295)	(0.314)	(0.159)	(0.455)	(0.487)	(0.176)	(0.288)	(0.260)
Ln FDI	0.018 *	−0.018	0.000	0.067 ***	0.153 ***	0.220 ***	0.016	0.145 ***	0.161 ***
	(0.011)	(0.029)	(0.034)	(0.016)	(0.041)	(0.049)	(0.014)	(0.028)	(0.035)
Ln OPEN	0.039 **	−0.239 ***	−0.200 ***	−0.048 **	−0.360 ***	−0.409 ***	0.020	−0.039	−0.019
	(0.016)	(0.043)	(0.051)	(0.022)	(0.057)	(0.068)	(0.016)	(0.033)	(0.034)
Ln R&D	−0.045 ***	0.079 ***	0.035	−0.006	0.152 ***	0.146 ***	−0.048 ***	−0.092 **	−0.140 ***
	(0.008)	(0.026)	(0.030)	(0.011)	(0.036)	(0.043)	(0.015)	(0.038)	(0.039)
Ln ENG	1.205 ***	−0.394 ***	0.811 ***	1.175 ***	−0.034	1.141 ***	1.059 ***	−0.520 ***	0.539 ***
	(0.038)	(0.112)	(0.129)	(0.053)	(0.202)	(0.043)	(0.068)	(0.113)	(0.134)
Ln ER	0.074 ***	0.119 ***	0.193 ***	0.074 ***	0.139 **	0.213 ***	0.010	−0.021	−0.011
	(0.013)	(0.036)	(0.042)	(0.020)	(0.054)	(0.067)	(0.013)	(0.027)	(0.028)

Notes: Robust standard errors are in parentheses. The significance levels 10%, 5%, and 1% are noted by *, **, and ***, respectively.

**Table 7 entropy-24-01042-t007:** Robustness test for eliminating outliers.

Variable	China			Northern China			Southern China		
	Dependent Variable: Ln CO_2_			Dependent Variable: Ln CO_2_			Dependent Variable: Ln CO_2_		
	Direct Effect	Indirect Effect	Total Effect	Direct Effect	Indirect Effect	Total Effect	Direct Effect	Indirect Effect	Total Effect
Ln GF	−0.004	−0.111 ***	−0.115 **	−0.001	−0.026	−0.027	−0.022	−0.027	−0.049 *
	(0.016)	(0.039)	(0.050)	(0.025)	(0.060)	(0.076)	(0.016)	(0.022)	(0.029)
Ln PGDP	−0.257 ***	0.267 *	0.010	0.378 ***	−0.391	−0.014	−0.432 ***	0.689 ***	0.257 ***
	(0.044)	(0.117)	(0.121)	(0.099)	(0.322)	(0.372)	(0.096)	(0.097)	(0.097)
Ln POP	0.110 ***	−0.066	0.044	0.027	−0.205 **	−0.179	0.075 **	−0.085	−0.010
	(0.020)	(0.050)	(0.059)	(0.032)	(0.096)	(0.111)	(0.038)	(0.130)	(0.151)
Ln IND	0.068	−0.235 *	−0.167	0.245 *	−0.471	−0.226	0.173 ***	−0.333 **	−0.160
	(0.048)	(0.130)	(0.150)	(0.125)	(0.353)	(0.455)	(0.061)	(0.168)	(0.160)
Ln URB	−0.010	1.301 ***	1.292 ***	−0.668 ***	1.760 ***	1.092 **	0.158	−1.075 ***	−0.917 ***
	(0.102)	(0.294)	(0.312)	(0.162)	(0.480)	(0.517)	(0.175)	(0.289)	(0.259)
Ln FDI	0.020 *	−0.013	0.007	0.068 ***	0.158 ***	0.226 ***	0.017	0.143 ***	0.160 ***
	(0.010)	(0.029)	(0.034)	(0.017)	(0.044)	(0.052)	(0.014)	(0.029)	(0.035)
Ln OPEN	0.038 **	−0.237 ***	−0.199 ***	−0.049 **	−0.363 ***	−0.412 ***	0.019	−0.039	−0.020
	(0.015)	(0.041)	(0.048)	(0;.023)	(0.060)	(0.072)	(0.016)	(0.033)	(0.033)
Ln R&D	−0.045 ***	0.078 ***	0.033	−0.005	0.158 ***	0.153 ***	−0.050 ***	−0.090 **	0.140 ***
	(0.008)	(0.025)	(0.028)	(0.012)	(0.038)	(0.045)	(0.015)	(0.038)	(0.040)
Ln ENG	1.202 ***	−0.375 ***	0.826 ***	1.175 ***	−0.034	1.141 ***	1.055 ***	−0.510 ***	0.545 ***
	(0.038)	(0.112)	(0.128)	(0.055)	(0.212)	(0.245)	(0.068)	(0.114)	(0.134)
Ln ER	0.069 ***	0.119 ***	0.188 ***	0.072 ***	0.142 **	0.214 ***	0.011	−0.022	−0.011
	(0.013)	(0.036)	(0.041)	(0.020)	(0.057)	(0.071)	(0.013)	(0.027)	(0.028)

Notes: Robust standard errors are in parentheses. The significance levels 10%, 5%, and 1% are noted by *, **, and ***, respectively.

## Data Availability

The data is available from the corresponding authors of this paper.

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
