# Peer review of "The Spatial Heterogeneity Effect of Green Finance Development on Carbon Emissions"

_entropy, 2022, doi:10.3390/e24081042_

Round 1

Reviewer 1 Report

The paper has certain research value, but there are still some problems, and the following modifications need to be made before the paper is accepted:  

1. The whole paper is made up of tables, and there is only one graph, which can show the results more intuitively. Too many watches make the article look boring.  

2. Please provide the corresponding chart making software and statistical analysis software in the article.  

3, the name of the table is too short to look too perfunctory, please add.  

4. The discussion and conclusion should be described separately.  Don't mix.

Author Response

Dear reviewers:

We sincerely appreciate the honorable reviewer of the Entropy for your careful review and comments. We have made some revisions in response to the suggestions. We have carefully revised the textual expression of the paper one by one, including the references. We have marked changes made in a DIFFERENT COLOR to facilitate the next round of review. Changes in words and additions to sentences are marked in red. The following is a response to the review comments, and we would like to invite the editors and reviewer to review them.

Reviewer's Comments to Author:

Reviewer: 1

1. The whole paper is made up of tables, and there is only one graph, which can show the results more intuitively. Too many watches make the article look boring.

Respond: Thank you very much for your hard work and valuable advice. Your suggestion is very helpful for improving the quality of the article.

First of all, according to your suggestion, we delete the multiple tables and describe the content in the original table in detail through the text. The newly added content is shown in lines 247-275 in the manuscript.

Secondly, we add a description of the changes in green credit, green securities, green insurance, green investment and green finance indicators for 2008-2019. According to the data from 2008 to 2019, the changes of each indicator are shown, and Fig. 2 is drawn in order to show the results of the article more intuitively, make the article look more interesting and read more convenient. Figure 2 is shown in line 239 of the manuscript, and the details are shown in lines 230-238.

2. Please provide the corresponding chart making software and statistical analysis software in the article.

Respond: Thank you for your hard work and valuable advice. Your valuable suggestions make this paper more readable. The software used for taking pictures in the manuscript is Origin2018. The software used for statistical analysis was Stata16.0. Details have been added to the manuscript, as detailed in notes 1, line 133, and line 445.

3. The name of the table is too short to look too perfunctory, please add.

Respond: Thank you for your dedicated work. According to your valuable opinions, we will further explain the shorter table name.

Revise "Descriptive statistics of variables" to descriptive statistics of variables analysis results, 2008-2019".

Revise "Robustness checks" to "Robustness test for substitution of explained variables".

The manuscript has been modified in lines 301 and line 428.

4. The discussion and conclusion should be described separately.Don't mix.

Respond: Thank you for your hard work and valuable advice. Your suggestion makes this article clear. We elaborate the conclusions and policy recommendations separately based on your recommendations. The contents are as follows:

1) Conclusions

This paper adopts panel data of China's 30 provinces from 2008 to 2019 and constructs the green finance index from four perspectives, i.e., green credit, green securities, green insurance, and green investment. This paper uses Stata16.0 software to test the regression of SDM model to evaluate the spatial effect of green finance on carbon emissions. We further analyze the heterogeneous effect in the south and north of China. The empirical results obtained in this paper are as follows. Firstly, China's green financial development can significantly reduce carbon emissions, and regional differences are obvious. Secondly, in the southern region, the impact of green finance development on regional and adjacent carbon emissions is not obvious, but generally, green finance can significantly reduce carbon emissions. In the northern region, the impact of green finance on carbon emissions is not obvious. Finally, across the country, the development of green finance has an inverted u-shaped relationship with carbon emissions in surrounding areas, and in general, the development of green finance in China has an inverted u-shaped relationship with carbon emissions.

2) Policy Recommendations

The development of green finance can effectively reduce carbon emissions, but there are certain differences in this effect in different regions. Therefore, the implementation of different green finance development policies is crucial for local contexts. Based on this, we propose the following policy recommendations.

First, in order to effectively reduce carbon emissions, the implementation scope of green finance should be expanded. Green finance can significantly reduce carbon emissions at the national level. The green finance of neighbouring provinces has a significant and negative effect on local carbon emissions, while the effect of local green finance on local carbon emissions is statistically insignificant. The indirect/ spillover effect is more prominent than the direct effect regarding carbon emission reduction. The total effect is very close to the indirect effect. It may be due to the fact that green finance in neighbouring provinces is implemented in a larger scope than in its own provinces. As a result, the larger the scope of green finance, the more carbon emissions are reduced. It indicates that green finance should be implemented nationwide and not be concentrated in one region. The effect of green finance development on carbon emissions in the whole country is first promoted and then decreased. In the initial stage of green finance development, green finance will promote carbon emissions. With the further development of green finance, green finance will reduce carbon emissions after reaching a certain threshold. There is an inverted U-shaped relationship between green finance in neighbouring regions and carbon emissions in the region, and the total effect of green finance on carbon emissions is also inverted U-shaped.

Second, in general, the development of green finance in the south and across China can significantly reduce carbon emissions, while the effect of green finance in the north is not significant. Compared with the south and the whole country, the effect of green finance on carbon emissions in the south and the whole country is not obvious. The development of green finance in the region has a significant spatial spillover effect on carbon emissions in adjacent areas, and there is no such spatial spillover effect in the southern region. Because of these substantial regional differences, in order to achieve the desired target, that is, emission peak and carbon neutrality, policies that stimulate green finance have to adjust to accommodate local conditions, such as heating supply, economic development level, green finance development level, the proportion of the secondary industry, population, and urbanization level.

Third, avoid "one size fits all" by implementing differentiated green finance development policies in different regions of North and South. Since green finance can effectively reduce carbon emissions and achieve the goal of emission peak and carbon neutrality, we need to make an effort to stimulate green finance development. The prospective development of green finance is promising. However, we still have to identify and solve the difficulties and challenges in the process. One of the important aspects of green finance is information disclosure. It is necessary to require listed firms to disclose relevant environmental information and green investment policies. In addition, it is important to coordinate the relationship between government and financial institutions and provide low-interest rates and low loan thresholds to financial institutions developing green finance.

The manuscript has been modified in lines 441-500.

Once again, thank you very much for your constructive comments and suggestions which would help us both in English and in depth to improve the quality of the paper. We look forward to hearing from you regarding our submission. We would be glad to respond to any further questions and comments that you may have.

Reviewer 2 Report

I think this paper is interesting. i have some concerns and suggestions for improvement .

1. based on the title I expected the paper would include both public and private firms.  But it only seems to be public firms.  What about venture capital, crowdfunding, and other forms of finance?
 See for example 

  • Bianchini, Roberto, & Annalisa Croce, 2022. "The Role of Environmental Policies in Promoting Venture Capital Investments in Cleantech Companies" Review of Corporate Finance

  • https://go.gale.com/ps/i.do?id=GALE%7CA259155182&sid=googleScholar&v=2.1&it=r&linkaccess=abs&issn=00355593&p=AONE&sw=w&userGroupName=anon%7E737ce30b

  •  

https://www.sciencedirect.com/science/article/pii/S1057521916300011?casa_token=z-PJeDyRL-UAAAAA:v2XE1E7RchrbJTbGh83pzLC6bZ4qlzDSYamqi0vW6Nh_XyfIjL7oQWO-cZGLUYoS6x-BdUbWlVPQ

https://www.sciencedirect.com/science/article/pii/S0959652622025148?casa_token=-07m-TMeJ9sAAAAA:VyMUdcRJgOGaowjlzxr4iVkcOhJxcNzXieawiyhBef9PYx-zJLM9alEgL3Mz2UtZMLyym2VtcKzj

2. I wonder about survivorship bias in the data.  Did some firms go bankrupt or become merged?

3. were there any pertinent regulatory changes over the sampl years?

4. the paper does not discuss limitations and future research.  Perhaps you could include a section, which might include for instance the issue of private firms (venture capital and crowdfunding) as mentioned in point 1 above.

5. are there any outliers driving results in the data?  Any issues of missing variables or reverse causality?

I hope these comments are helpful. 

  •  
  •  

Author Response

Response to Reviewer2

Dear reviewer:

We sincerely appreciate the honorable reviewer of the Entropy for your careful review and comments. We have made some revisions in response to the suggestions. We have carefully revised the textual expression of the paper one by one, including the references. We have marked changes made in a DIFFERENT COLOR to facilitate the next round of review. Changes in words and additions to sentences are marked in red. The following is a response to the review comments, and we would like to invite the editors and reviewer to review them.

Reviewer's Comments to Author:

Reviewer: 2

1. Based on the title I expected the paper would include both public and private firms.  But it only seems to be public firms.  What about venture capital, crowdfunding, and other forms of finance?

Respond: Thank you for your hard work and valuable advice. Your valuable suggestions make this paper more readable. According to your suggestion, we draw on Bianchini, Roberto, & Annalisa Croce's research and consider venture capital from the perspective of private companies. And the corresponding empirical analysis and discussion, detailed content see in the manuscript 364-375 lines.

2. I wonder about survivorship bias in the data. Did some firms go bankrupt or become merged?

Respond: Thank you very much for your hard work. From 2008-2019, some companies were bankrupt or merged. The research object of this paper is to select companies that exist annually (excluding exit) as samples and do not track bankrupt enterprises. In addition, to the annexed enterprises, this paper only considers the annexed enterprises.

3. Were there any pertinent regulatory changes over the sample years?

Respond: Thank you for your hard work and valuable advice. Your valuable suggestions make this paper more readable. Since the industrial Bank put forward the "Equator Principles", China has formulated a series of green finance development policies, most of which revolve around the theme of "developing green finance to help achieve carbon emission reduction targets". This makes this paper more valuable for research. The details have been added to note â‘¢ of the manuscript.

4. The paper does not discuss limitations and future research. Perhaps you could include a section, which might include for instance the issue of private firms (venture capital and crowdfunding) as mentioned in point 1 above.

Respond: Thank you very much for your hard work and valuable opinions. Your suggestion provides a new perspective for this study. According to your suggestion, we comprehensively consider the impact of venture capital on carbon emissions from the perspective of private companies and conduct relevant empirical tests. The empirical test results are shown in Table 1. The indirect effect of venture capital on carbon emissions is obvious at the level of 10%, and the direct effect and total effect of venture capital on carbon emissions are not obvious. This shows that increasing risk investment in surrounding areas can reduce carbon emissions in the region. The reason may be that the implementation scope of surrounding areas is larger, and the implementation effect is greater. Therefore, in order to further promote the emission reduction effect of venture capital on carbon emissions, the scope of implementation of venture capital should be expanded, and the intensity of venture capital should be increased.     See lines 354-365 in the manuscript for details. 

Table 1. Three impacts of venture capital on China's carbon emissions

Variable

Direct 

effect

Indirect effect

Total

effect

Variable

Direct

effect

Indirect

effect

Total

effect

Ln VC

0.005

-0.018*

-0.012

Ln FDI

0.013

0.004

0.016

(0.004)

(0.010)

(0.012)

(0.011)

(0.029)

(0.035)

Ln PGDP

-0.270***

0.305**

0.035

Ln OPEN

0.033**

-0.223***

-0.190***

(0.044)

(0.122)

(0.126)

(0.016)

(0.042)

(0.050)

Ln POP

0.110***

-0.049

0.060

Ln R&D

-0.042***

0.070***

0.032

(0.020)

(0.052)

(0.061)

(0.008)

(0.026)

(0.030)

Ln IND

0.105**

-0.046

0.059

Ln ENG

1.211***

-0.378***

0.833***

(0.043)

(0.116)

(0.136)

(0.038)

(0.114)

(0.130)

Ln URB

0.032

1.080***

1.111***

Ln ER

0.067***

0.094***

0.160***

(0.101)

(0.294)

(0.318)

(0.013)

(0.035)

(0.039)

5. Are there any outliers driving results in the data? Any issues of missing variables or reverse causality?

Respond: Thank you for your hard work and valuable comments. Your comments help improve the quality of this article.

First, the abnormal values in the sample may lead to abnormal regression results. In order to avoid the impact of outliers on the regression results, this paper further eliminates the lowest and highest 1 % of sample data in carbon emissions and green finance. After excluding the outliers, the direction and visibility of green finance on carbon emission coefficient in China's national, northern and southern regions have not changed. Therefore, the regression conclusion is not affected by the sample outliers. For details, see lines 431-440 of the manuscript.

Second, the explained variable of this paper is carbon emissions, and the explanatory variable is green finance. According to the existing research, green finance can effectively affect carbon emissions, but as a kind of environmental pollution, carbon emissions have no obvious reverse causality to green finance. The existing research on the impact of green finance on environmental pollution, such as Li, C.G. and Gan, Y. (2020), Wang, X.X. et al. (2021) et al., have not carried out the endogenous test. Thus, there may be no obvious reverse causality between green finance and carbon emissions.

In addition, the fixed effect model and spatial econometric model used in this paper can solve endogeneity to a certain extent.

[1] Li, C.G.; Gan, Y. The spatial spillover effects of green finance on ecological environment-empirical research based on spatial econometric model. Environmental Science and Pollution Research. 2020, 28, 5651-5665.

[2] Wang, X.X.; Huang, J.Y.; Xiang, Z.M.; Huang, J.L. Nexus between green finance, energy efficiency, and carbon emission: Covid-19 implications from BRICS countries. Frontiers in Energy Research. 2021, 9.

Once again, thank you very much for your constructive comments and suggestions which would help us both in English and in depth to improve the quality of the paper. We look forward to hearing from you regarding our submission. We would be glad to respond to any further questions and comments that you may have.
